

# Effects of plant growth-promoting rhizobacteria on blueberry growth and rhizosphere soil microenvironment

Mengjiao Wang[1,2,3] and Xinlong Yang[1]

[1] School of Biological Science and Engineering, Shaanxi University of Technology, Hanzhong, Shaanxi, China
[2] Collaborative Innovation Center for Comprehensive Development of Biological Resources in Qinling-Ba Mountains, Hanzhong, Shaanxi, China
[3] Shaanxi Key Laboratory of Bioresources, Hanzhong, Shaanxi, China

Corresponding author
Mengjiao Wang,
amy133253@126.com

## ABSTRACT

**Background:** Plant growth-promoting rhizobacteria (PGPR) have a specific symbiotic relationship with plants and rhizosphere soil. The purpose of this study was to evaluate the effects of PGPR on blueberry plant growth, rhizospheric soil nutrients and the microbial community.

**Methods:** In this study, nine PGPR strains, belonging to the genera *Pseudomonas and Buttiauxella*, were selected and added into the soil in which the blueberry cuttings were planted. All the physiological indexes of the cuttings and all rhizospheric soil element contents were determined on day 6 after the quartic root irrigation experiments were completed. The microbial diversity in the soil was determined using high-throughput amplicon sequencing technology. The correlations between phosphorus solubilization, the auxin production of PGPR strains, and the physiological indexes of blueberry plants, and the correlation between rhizospheric microbial diversity and soil element contents were determined using the Pearson's correlation, Kendall's tau correlation and Spearman's rank correlation analysis methods.

**Results:** The branch number, leaf number, chlorophyllcontentand plant height of the treated blueberry group were significantly higher than those of the control group. The rhizospheric soil element contents also increased after PGPR root irrigation. The rhizospheric microbial community structure changed significantly under the PGPR of root irrigation. The dominant phyla, except Actinomycetota, in the soil samples had the greatest correlation with phosphorus solubilization and the auxin production of PGPR strains. The branch number, leaf number, and chlorophyllcontent had a positive correlation with the phosphorus solubilization and auxin production of PGPR strains and soil element contents. In conclusion, plant growth could be promoted by the root irrigation of PGPR to improve rhizospheric soil nutrients and the microenvironment, with modification of the rhizospheric soil microbial community.

**Discussion:** Plant growth could be promoted by the root irrigation of PGPR to improve rhizospheric soil nutrients and the microenvironment, with the modification of the rhizospheric soil microbial community. These data may help us to better understand the positive effects of PGPR on blueberry growth and the

![PeerJ]

rhizosphere soil microenvironment, as well as provide a research basis for the subsequent development of a rhizosphere-promoting microbial fertilizer.

## INTRODUCTION

Blueberries are a popular fruit and have potential health benefits in the prevention of common chronic diseases (*Tobar-Bolaños et al., 2021*). Similar to all plants, blueberry is profoundly influenced by environmental factors in its rhizosphere soil, such as nutritional elements, rhizospheric microorganisms and pH (*Yurgel et al., 2019*; *Yang et al., 2022*). Improvements in the rhizospheric microecological environment have positive effects on crop productivity and sustainable development (*Ren et al., 2021*). The plant growth-promoting rhizobacteria (PGPR), important rhizospheric microorganisms, play a critical role in promoting plant health and regulating the soil microecological environment (*Santoyo et al., 2021*).

PGPR have specific symbiotic relationships with plants and positively affect plant life cycles in direct and indirect manners (*Singh et al., 2015*; *Nagrale et al., 2023*). PGPR directly promote plant growth by enhancing the acquisition of soil nutrients, nitrogen fixation and the mobilization of key nutrients (phosphorus, potassium and iron) (*Rashid et al., 2016*). PGPR inhabit the rhizosphere and develop nodules on legumes and endophytes that can colonize the interior tissues of plants (*Meena et al., 2020*). As biocontrol agents, PGPR are readily available and experience less adverse effects from the various stresses that plants encounter (*Hafez et al., 2021*). Thus, they can be an efficient economical tool for increasing the productivity of important agriculture crops (*Calvo, Nelson & Kloepper, 2014*; *Vaishnav et al., 2022*). Furthermore, the application of PGPR has the potential to regulate the microecological environment in the rhizosphere through the elicitation of several physiological and molecular mechanisms (*Shi et al., 2022*). For example, PGPR strains can regulate the microecological environment through the production of exopolysaccharides and ACC deaminase, the accumulation of various osmolytes (such as proline, sugars, amino acids, polyamines and betaines) and antioxidant alteration (*Tang et al., 2020*; *Zhao et al., 2023*). PGPR strains can also improve root systems, including antioxidant capability, the production of exopolysaccharides (EPS) and siderophores, modulation of phytohormones (such as ABA, IAA and ethylene), synthesis of osmolytes, uptake of minerals and control of phytopathogens (*De Andrade et al., 2023*; *Arora et al., 2018*). Several PGPR strains have been reported to increase soil organic matter, and to improve soil structure and water retention capacity as bioinoculants (*Arora et al., 2020*). Our previous study found that a poor soil ecosystem could be restored *via* the bioremediation method, by intervening with soil bacterial diversity and stability using PGPR (*Wang et al., 2019*).

In this study, nine PGPR were applied to the blueberry plants *via* root irrigation. The physiological indexes of blueberry plants were evaluated on day 6, after the end of

quartic root irrigation experiments. The soil element contents and microbial diversity in the rhizosphere were measured. The correlations between the physiological indexes of the blueberry plants with rhizospheric microenvironmental factors, soil element contents and microbial diversity were determined. This study could represent an initial step in developing efficient and environmentally friendly PGPR fertilizer to promote blueberry plant growth.

## MATERIALS AND METHODS

### Plant root irrigation

Nine PGPR strains, as selected in the article of *Wang, Sun & Xu (2022)*, were chosen for the root irrigation of high phosphorus and silicate solubilization, auxin production and nitrogen fixation capabilities. These nine PGPR strains belong to the *Pseudomonas and Buttiauxella* genera. The concentrations of phosphorus in the supernatant, which was collected from a liquid medium inoculated with these nine PGPR strains, were 0.35–4.99 mg/L. The concentration of auxin produced by the nine strains was more than 12.5 mg/L. The nine PGPR strains are shown in Appendix Table 1.

The 2-year-old branches of blueberry plants were shortened into cuttings using pruning shears. Cuttings with one or two leaves and one or two buds were chosen for immediate insertion into the soil. All cuttings were grown at 25 °C under continuous illumination (~1,500 Lx). Blueberry cuttings with five leaves and a 10 cm height were chosen for root irrigation. The experiment used 18 cm height pots with soil samples collected from a blueberry field. All the PGPR strains were incubated in 5 mL of liquid beef extract peptone medium individually and incubated at 28 °C for 2 days. Every liquid beef extract peptone medium with a high cell density (Optical density (OD) (600) = 0.8) was diluted using sterilized water to a final volume of 50 mL. Twenty cuttings were irrigated every 6 days (four times in total) with prepared liquid beef extract peptone medium. For the control group (CK), another 5 mL of sterilized liquid beef extract peptone medium was diluted using sterilized water to a final volume of 50 mL. In addition, the 50 mL diluted liquid medium was irrigated to another twenty cuttings every 6 days for a total of four irrigations. All plants were grown at 25 °C under continuous illumination (~1,500 Lx). Rhizosphere soil samples were collected from the roots of the cuttings on day 6 after the end of the quartic root irrigation experiment. The above-mentioned plant root irrigation experiments were repeated three times.

### Determination of blueberry plant physiological index and soil sample collection

The physiological indexes of the cutting-seedlings were determined on day 6 following the quartic root irrigation experiments. The eighth leaf of every cutting-seedling was harvested, weighed and finely ground in liquid $N_2$. Total chlorophyll (Chl) was extracted with 95% ethanol, and chlorophyll concentrations were calculated according to the *Lichtenthaler (1987)* method. After the branch and leaf number were counted, the rhizosphere soil samples were collected using a method described in a previous study (*Fujii, Akihiro & Syuntro, 2004*). The soil samples collected in each treatment were fully

mixed and then stored at 4 °C before use. Then, all cutting-seedlings were carefully removed from soil and washed in distilled water until there was no excess soil attached to the roots. Primary root length was evaluated on plant images using Image J software (NIH) (*Kohanová et al., 2018*). Plant height was measured as the distance from the base of the plant to the tip of the main shoot (*Kaur et al., 2021*).

## Determination of soil element contents and analysis of microbe DNA sequences in soil sample

The organic carbon content (OCC), total nitrogen content (TNC), total phosphorous content (TPHC), total potassium content (TPOC), hydrolysable nitrogen content (HNC), available phosphorous content (APHC) and available potassium content (APOC) in soil samples were determined using the methods reported in *Wang et al. (2021)*.

The genomes of microbes in soil samples were extracted using a DNA extraction kit (Fast DNA Spin Kit for Soil, MP Biomedicals, Santa Ana, CA, USA). These Hiseq sequencing results in double-ended sequence data (pairwise. FASTQ files) were submitted to the Sequence Read Archive (https://submit.ncbi.nlm.nih.gov/subs/sra/), and the submission number was obtained. All the analysis, including the amplification and purification of 16S rRNA genes and ITS genes, library preparation and sequencing, and data analysis, were carried out using the same method as described in *Wang, Sun & Xu (2022)*.

## Data analysis

All experiments were repeated in triplicate. The physiological indexes of the cutting-seedlings, and soil element contents were expressed using mean and standard derivation. They were tested for statistical distribution before ANOVA analysis with a significant difference between two data at $p < 0.05$ or $p < 0.01$, where bars with different letters indicate a significant difference between the data. The distribution of microorganisms with a relative abundance greater than or equal to 1% in blueberry cutting-seedlings rhizospheres was expressed using the mean value of three parallel experiments. Significant differences between different species were determined *via* linear discriminant analysis (LDA) effect size (LEfSe) (https://github.com/biobakery/galaxy_lefse) with two as the default setting filter value for the LDA score. Pearson's correlation coefficient, Kendall's tau correlation coefficient and Spearman's rank correlation coefficient were analyzed using a multivariate process of the GLM in SPSS (Statistical Product and Service Solutions) software to identify and quantify the nature of the link between the phosphorus solubilizing ability and auxin production ability of PGPR strains with rhizosphere soil microbial diversity, soil element content and plant growth status. The correlations between the physiological indexes of the blueberry plants with rhizosphere microenvironmental factors, and between rhizosphere microbial diversity and rhizosphere soil element contents were also analyzed *via* a multivariate process of the GLM in SPSS software (*Cornbleet & Shea, 1978*; *Mangena, 2021*; *Wang et al., 2021*). Principal component analysis (PCA) was conducted using the SPSS v23.0 software (*Zhou et al., 2021*) to obtain more information about the nature of the link between phosphorus solubilizing ability and the auxin

production ability of PGPR strains with rhizosphere microenvironmental factors and plant growth status.

## RESULTS

### Effects of PGPR strains on blueberry plant growth and element content of blueberry rhizosphere soil

The physiological indexes of the blueberry plants in control and treatment groups are shown in Fig. 1. The PGPR treatment groups had a greater number of branches and a greater plant height of cutting-seedlings (Figs. 1A, 1D). All the strain treatments, except B5 and B7, had higher leaf numbers compared with the control (Fig. 1B). Among them, the number of leaves was the highest in the cutting-seedlings treated with strain B8, which was 32.5% more than the control (Fig. 1B). Strains B2, B3, B6 and B9 significantly increased the Chl concentration of the eighth leaf of the blueberry cutting-seedlings by 20.0%, 43.4%, 36.1% and 37.2%, respectively (Fig. 1C). The root lengths in treatments B3, B6, and B9 were 13.3, 11.9 and 11.0 cm, respectively, while it was 9.5 cm in the control group (Fig. 1D).

The concentrations of major environmental elements, OCC, TNC, HNC, TPHC, APHC, TPOC and APOC, were 433.03–583.50 g/kg, 21.50–29.26 g/kg, 5.75–10.73 g/kg, 224.32–445.13 mg/kg, 129.35–174.12 mg/kg, 1.21–4.58 g/kg and 0.45–0.23 g/kg, respectively (Fig. 2). Inoculation with the strains significantly increased OCC in the rhizosphere soil compared with the control (Fig. 2A). All strain isolates except B7 increased the TNC in rhizosphere soil. Strains B1, B2, B3, B6 and B7 significantly increased the HNC in rhizosphere soil samples of blueberry cutting-seedlings by 37.1%, 50.1%, 63.6%, 86.8% and 59.8%, respectively (Fig. 2B). All strain isolates, except B4 and B5, increased the TPHC in the rhizosphere soil, and strain B3 presented the highest level of APHC in rhizosphere soil samples with 34.9% more than the control (Fig. 2C). The TPOC in rhizosphere soil samples that were collected from the cutting-seedlings inoculated with the strains, except for strains B4, B5 and B6, was greater than in the control (Fig. 2D). All strain treatments significantly increased APOC in the rhizosphere soils except for strain B7 (Fig. 2D).

### Effects of PGPR strains on microbial community structure of blueberry rhizosphere soil

There were 1,145 species of bacterial genera in the rhizosphere soil samples gathered from the cutting-seedlings. Among them, 18 genera with a relative abundance greater than 1% were identified: *Granulicella*, *Occallatibacter*, *Solibacter*, *Acidothermus*, *Bryobacter*, *Mucilaginibacter*, *Bauldia*, *Bradyrhizobium*, *Paraburkholderia*, *Buttiauxella*, *Devosia*, *Dongia*, *Haliangium*, *Pseudolabrys*, *Pseudomonas*, *Sphingomonas*, *Lacunisphaera* and *Opitutus* (Fig. 3A). These genera belonged to Acidobacteriota, Actinomycetota, Bacteroidota, Pseudomonadota and Verrucomicrobiota phyla (Fig. 3A).

There were significant differences in bacterial diversity between the rhizosphere samples collected from blueberry cutting-seedlings with different treatments. All strain isolates had a higher percentage of *Occallatibacter* and *Pseudomonas* than the control (Fig. 3A). The *Occallatibacter* in control soil samples was 5.27%, while the highest percentage of

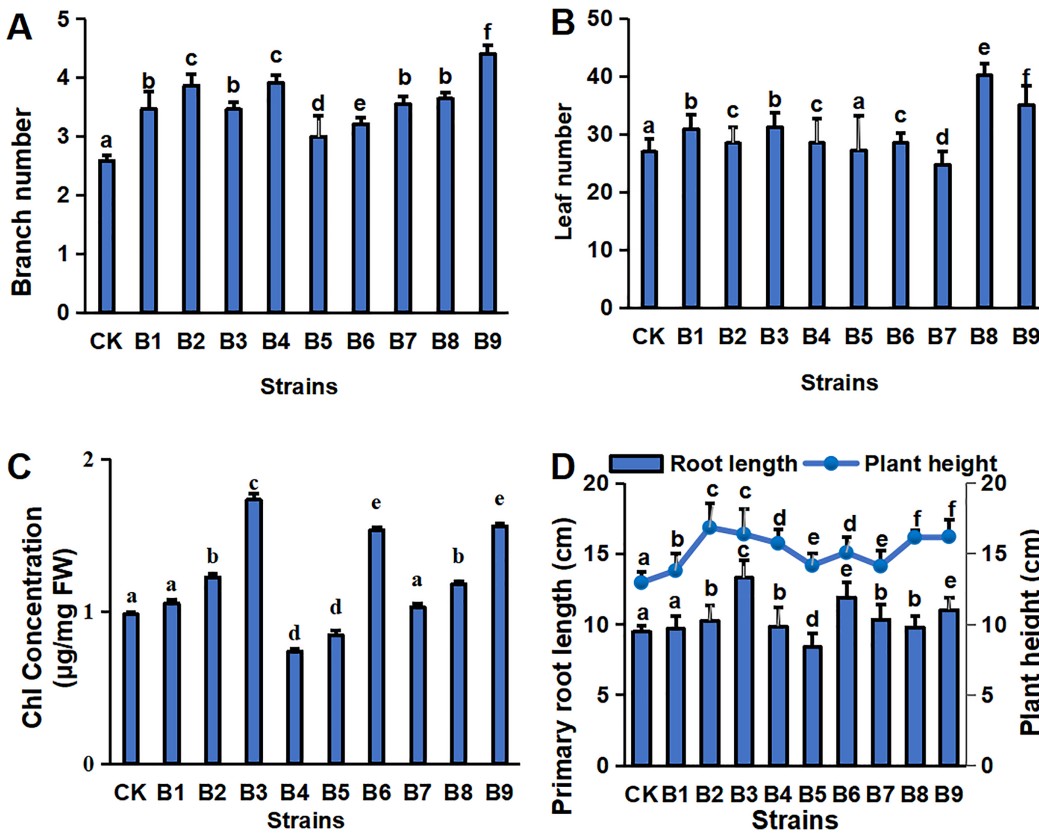

**Figure 1 Plant physiology in blueberry cutting-seedlings rhizospheres.** Numbers of branches (A) and leaves (B), chl (Chlorophyll) concentration (C), primary root length, and plant height (D) in blueberry cutting-seedlings rhizospheres. Bars with different letters indicate a significant difference between the data ($p < 0.05$).

*Occallatibacter* in strain B6-inoculated cutting-seedling rhizosphere samples was 12.53% (Fig. 3A). The highest percentage of *Pseudomonas* was 23.96% in the cutting-seedlings rhizosphere samples inoculated with the strains, while in the control soil sample it was only 4.43% (Fig. 3A). Compared with the percentage of *Devosia* (1.09%) and *Haliangium* (1.57%) in the control soil samples, the percentages of *Devosia* and *Haliangium* were lower than 1% in all rhizosphere samples treated with the strains (Fig. 3A). Additionally, treatment with different strains had different effects on bacterial diversity in the cutting-seedlings rhizosphere soil. The percentage of *Buttiauxella* in cutting-seedlings rhizosphere soil samples inoculated with *Buttiauxella* was significantly higher than in other treatments (Fig. 3A).

For fungal communities, most of the OTUs were classified as Ascomycota, Basidiomycota and Mucoromycota at the phylum level (Fig. 3B). On a genus level, the communities were dominated by *Ascitendus*, *Pezoloma*, *Rhexodenticula*, *Sarocladium*, *Thysanorea*, *Clitopilus*, *Gymnopilus*, *Rhizoctonia*, *Sebacina*, *Sistotrema*, *Sistotremella* and *Mortierella* in the blueberry cutting-seedlings rhizosphere (Fig. 3B). Significant differences in fungal diversity between the rhizosphere samples collected from blueberry cutting-seedlings with different treatments were also found. The percentages of *Ascitendus*

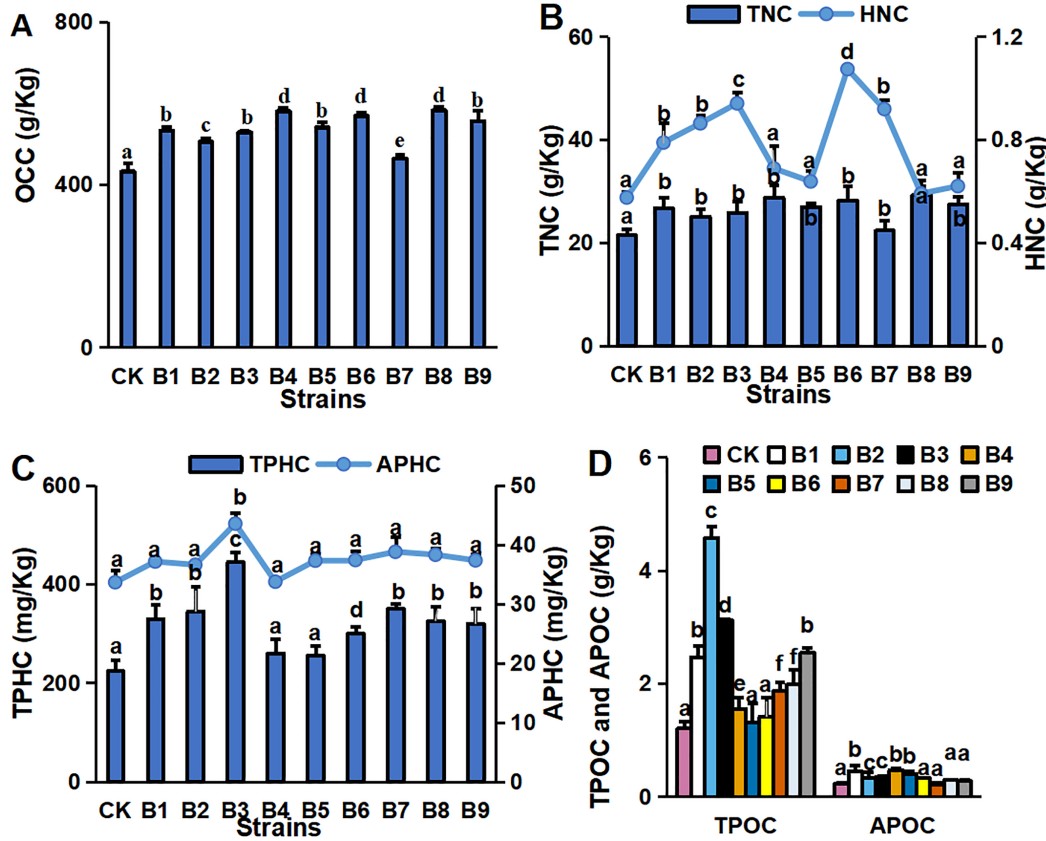

**Figure 2 Nutritive element contents in blueberry cutting-seedlings rhizospheres.** Organic carbon content (OCC) (A), total nitrogen content (TNC) (B), hydrolysable nitrogen content (HNC) (B), total phosphorous content (TPHC) (C), available phosphorous content (APHC) (C), total potassium content (TPOC) (D), and available potassium content (APOC) (D) in blueberry cutting-seedlings rhizospheres. Bars with different letters indicate a significant difference between the data ($p < 0.05$).

and *Thysanorea* in cutting-seedlings rhizosphere soil samples inoculated with the strain were significantly higher than in the control soil samples (Fig. 3B). Compared with the percentage of *Rhexodenticula* (2.29%), *Clitopilus* (15.36%) and *Sebacina* (49.50%) in the control soil samples, the percentages of *Rhexodenticula*, *Clitopilus* and *Sebacina* were significantly lower in the rhizosphere soil samples across all treatment groups (Fig. 3B). Different effects on fungal diversity were also found in rhizosphere samples of cutting-seedlings inoculated with different strains. The *Gymnopilus*, *Rhizoctonia*, *Sistotrema*, *Sistotremella* and *Mortierella* in rhizosphere soil were significantly enriched following inoculation with *Pseudomonas* isolates (Fig. 3B).

A total of 36 distinct bacterial biomarkers were identified using an LDA threshold score of ≥2.0. Inoculation with *Pseudomonas* isolate enriched phylotypes belonging to Actinobacteriota (Actinobacteria), Proteobacteria, Myxococcota and Acidobacteriota (Vicinamibacteria) (Fig. 4A). In addition, the total number of bacterial biomarkers in the soil samples collected from cutting-seedlings inoculated with *Pseudomonas* isolates was higher than that in the soil samples collected from cutting-seedlings with other treatments (Fig. 4A). The bacteria in cutting-seedlings inoculated with *Buttiauxella* isolates

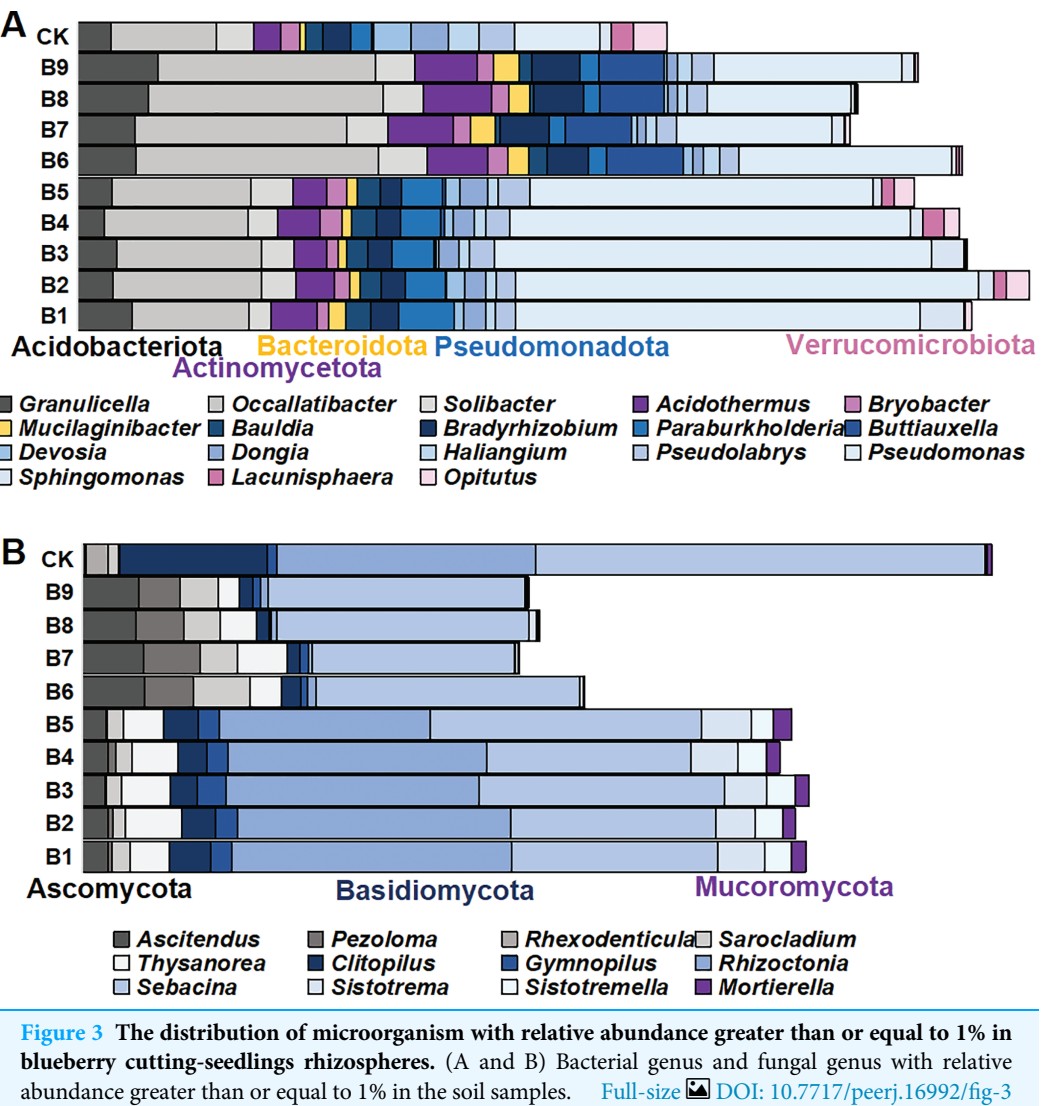

**Figure 3** **The distribution of microorganism with relative abundance greater than or equal to 1% in blueberry cutting-seedlings rhizospheres.** (A and B) Bacterial genus and fungal genus with relative abundance greater than or equal to 1% in the soil samples.

rhizosphere soil were abundant with Actinobacteria (Acidimicrobiia) and Firmicutes (Clostridia) (Fig. 4A). The specific phylotypes in the control group were taxonomically diverse and included members of Myxococcota (Haliangiales), Proteobacteria (Alphaproteobacteria) and Verrucomicrobiota (Opitutaceae) (Fig. 4A).

The analysis of fungal communities revealed 39 distinct biomarkers that were unevenly distributed among the microorganisms in the blueberry cutting-seedlings rhizospheres (Fig. 4B). The rhizosphere fungi of cutting-seedlings inoculated with Pseudomonas isolates were rich in diverse Ascomycota and Basidiomycota (Agaricomycetes) (Fig. 4B). In contrast, the specific fungi in inoculation treatment with Buttiauxella isolates were rich in Ascomycota, Basidiomycota and Mucoromycota (Fig. 4B). There were only seven distinct biomarkers that were differentially distributed in the rhizosphere samples from cutting-seedlings inoculated with Pseudomonas compared with the rhizospheres in other treatments (Fig. 4B).

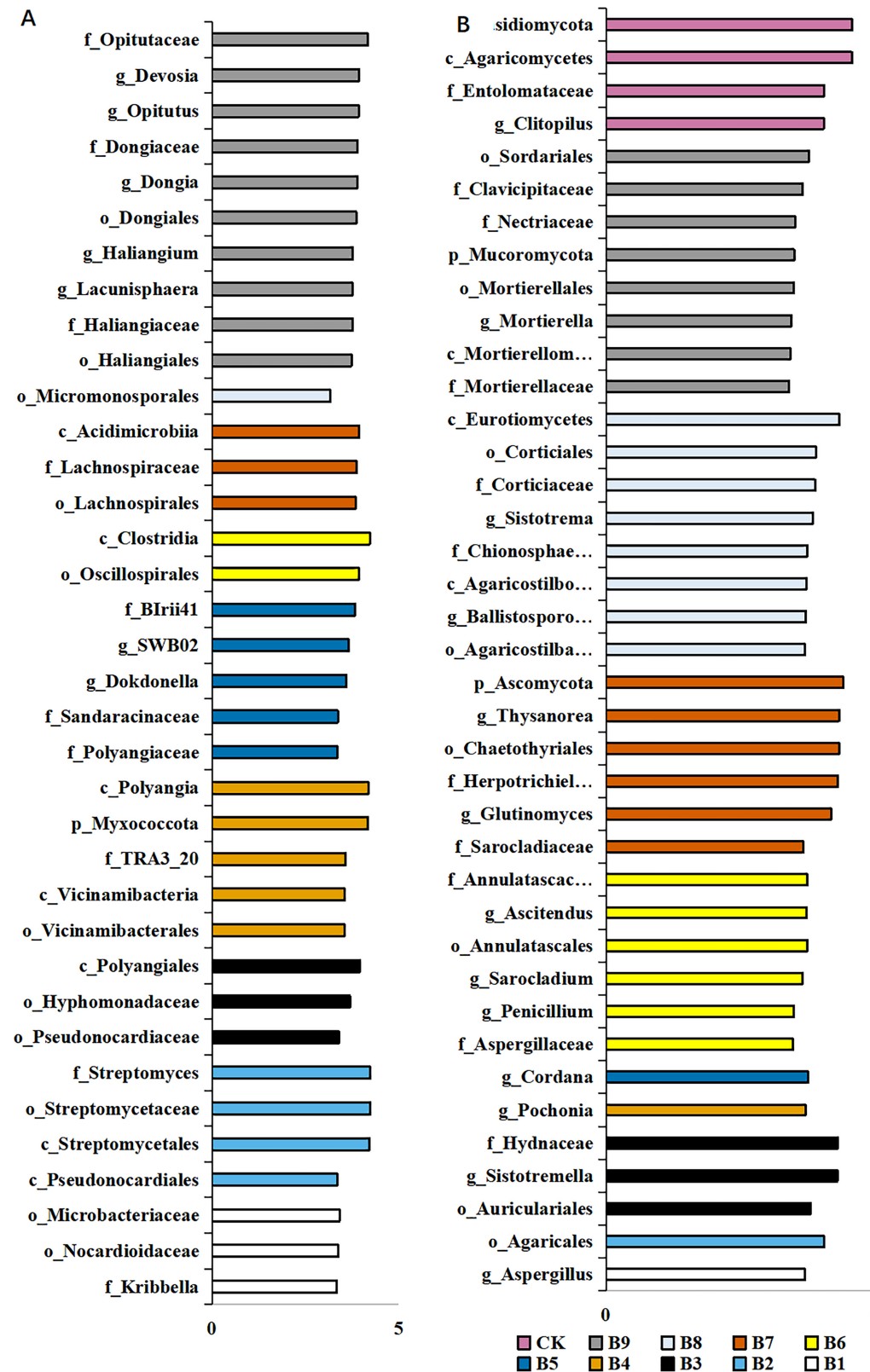

**Figure 4 LEfSe analysis of differentially abundant classes, orders, families and genera of microorganism in blueberry cutting-seedlings rhizospheres.** (A) The result of bacteria LEfSe analysis of differentially abundant and (B) result of fungi LEfSe analysis of differentially abundant. The LDA threshold score in the figure was equal or greater than 2.0.

**Table 1 Pearson's correlation analysis of phosphorus solubilizing ability and auxin production ability of PGPR strains with rhizosphere soil microbial diversity, soil element content and plant growth status.**

| Capacity of strains | Acidobacteriota | Actinomycetota | Bacteroidota | Pseudomonadota | Verrucomicrobiota | Ascomycota | Basidiomycota | Mucoromycota |
|---|---|---|---|---|---|---|---|---|
| Phosphorus | 0.967** | 0.044 | 0.923** | −0.579** | −0.698* | 0.881** | −0.908** | −0.792** |
| Auxin | 0.761** | 0.047 | 0.818** | −0.499** | −0.618** | 0.745** | −0.841** | −0.556** |
| | OCC | TNC | HNC | TPHC | APHC | TPOC | APOC | – |
| Phosphorus | 0.343 | 0.308 | 0.104 | 0.112 | −0.078 | −0.144 | −0.395* | – |
| Auxin | 0.201 | 0.152 | −0.124 | 0.228 | −0.053 | −0.011 | −0.390* | – |
| | Branch number | Leaf number | Chl | Primary root length | Plant height | – | – | – |
| Phosphorus | 0.387* | 0.457* | 0.425* | 0.244 | 0.208 | – | – | – |
| Auxin | 0.487** | 0.456* | 0.123 | −0.019 | 0.239 | – | – | – |

Notes:
Phosphorus, Phosphorus-solubilizing capacity of strains used for root irrigation; Auxin, Auxin production capacity of strains used for root irrigation; OCC, organic carbon content; TNC, total nitrogen content; TPHC, total phosphorous content; TPOC, total potassium content; HNC, hydrolysable nitrogen content; APHC, available phosphorous content; APOC, available potassium content. The underline entries means there were highly significant correlation between the two data.
* $p < 0.05$.
** $p < 0.01$.

## Correlations between the strains with rhizosphere microenvironmental factors and physiological indexes of blueberry plants

The dominant phyla, except Actinomycetota, in the soil samples had the greatest correlation with the phosphorus solubilization and auxin production of PGPR strains (Table 1). The APOC was the only environmental factor that had a correlation with phosphorus solubilization and auxin production, while the other environmental factors did not show significant correlations (Table 1). There was a strong correlation between microbial community and plant branch number (Table 1). Similar results are shown in Appendix Tables 2 and 4 and are based on Kendall's tau correlation coefficient and Spearman's rank correlation coefficient analysis methods.

To further understand the relationship between rhizosphere microbial diversity, rhizosphere soil element contents and plant growth indicators of blueberry seedlings, Pearson's correlation coefficient was calculated. The results showed that the dominant Bacteroidota and Basidiomycota phyla had the greatest correlation with the branch number of cutting-seedlings, while the Verrucomicrobiota phyla had the greatest correlation with the leaf number, Chl and the primary root length of cutting-seedlings (Table 2). The Actinomycetota was the dominant phyla that had a correlation with TPOC, phosphorus solubilization and the auxin production of the strains. Pseudomonadota in the soil samples had the greatest correlation with TPOC and APOC, while Verrucomicrobiota showed a strong correlation with TPHC (Table 2). There were significant correlations between the rhizosphere soil element contents and plant growth indicators of blueberry seedlings. The rhizosphere soil element contents, except APOC, had a significant correlation with each plant growth index in blueberry seedlings (Table 2). Most environmental elements in this study, including OCC, TNC, HNC, TPHC, APHC and

Table 2 Pearson's correlation analysis of rhizosphere soil microbial diversity with plant growth status and soil element contents, Pearson's correlation correlation analysis of soil element contents with plant growth status.

| | Branch number | Leaf number | Chl | Primary root length | Plant height | OCC | TNC | HNC | TPHC | APHC | TPOC | APOC |
|---|---|---|---|---|---|---|---|---|---|---|---|---|
| Acidobacteriota | 0.372* | 0.450* | 0.446* | 0.299 | 0.314 | 0.386* | 0.331 | 0.191 | 0.179 | −0.047 | −0.096 | −0.406* |
| Actinomycetota | 0.368* | 0.077 | 0.041 | −0.102 | 0.117 | 0.009 | 0.046 | 0.225 | 0.248 | −0.233 | 0.622** | 0.177 |
| Bacteroidota | 0.464** | 0.337 | 0.298 | 0.147 | 0.096 | 0.238 | 0.208 | 0.164 | 0.195 | −0.118 | −0.067 | −0.324 |
| Pseudomonadota | 0.233 | −0.167 | 0.029 | 0.157 | 0.341 | 0.258 | 0.207 | 0.316 | 0.339 | −0.136 | 0.559** | 0.716** |
| Verrucomicrobiota | −0.451* | −0.482** | −0.596** | −0.552** | −0.339 | −0.443* | −0.366* | −0.407* | −0.632** | −0.124 | −0.133 | 0.045 |
| Ascomycota | 0.381* | 0.220 | 0.322 | 0.303 | 0.267 | 0.318 | 0.255 | 0.400* | 0.243 | −0.122 | −0.071 | −0.327 |
| Basidiomycota | −0.515** | −0.348 | −0.320 | −0.252 | −0.347 | −0.412* | −0.338 | −0.264 | −0.262 | 0.151 | 0.033 | 0.274 |
| Mucoromycota | −0.087 | −0.246 | −0.295 | −0.215 | 0.024 | 0.102 | 0.087 | −0.058 | 0.038 | −0.081 | 0.242 | 0.709** |
| OCC | 0.477** | 0.574** | 0.168 | 0.123 | 0.559** | – | – | – | – | – | – | – |
| TNC | 0.389* | 0.523** | 0.090 | 0.044 | 0.431* | – | – | – | – | – | – | – |
| HNC | −0.011 | −0.380* | 0.432* | 0.635** | 0.162 | – | – | – | – | – | – | – |
| TPHC | 0.362* | 0.233 | 0.666** | 0.694** | 0.511** | – | – | – | – | – | – | – |
| APHC | −0.310 | 0.032 | 0.421* | 0.398* | −0.066 | – | – | – | – | – | – | – |
| TPOC | 0.509** | 0.165 | 0.417* | 0.333 | 0.652** | – | – | – | – | – | – | – |
| APOC | 0.132 | −0.010 | −0.291 | −0.143 | 0.108 | – | – | – | – | – | – | – |

Notes:
OCC, organic carbon content; TNC, total nitrogen content; TPHC, total phosphorous content; TPOC, total potassium content; HNC, hydrolysable nitrogen content; APHC, available phosphorous content; APOC, available potassium content. The underline entries means there were highly significant correlation between the two data.
* $p < 0.05$.
** $p < 0.01$.

TPOC, were significantly correlated with the plant growth indexes of blueberry seedlings (Appendix Tables 3 and 5). According to the results of PCA, there were five main principal components (Appendix Table 6). The absolute value of the eigenvector load was considered as a coefficient of correlation between the variable and principal component, where a larger value indicates a higher correlation between the variable and the principal component. In the first principal component, the phosphorus-solubilizing production of the strains (0.927), Acidobacteriota (0.927), Bacteroidota (0.888), Ascomycota (0.906) and Basidiomycota (−0.944) was a determining factor. Pseudomonadota (0.884) was a determining factor in the second principal component.

## DISCUSSION

### Stimulation of plant growth

PGPR are not only crucial in providing the soil with nutritional elements for plant growth, but they also restrict or inhibit the growth of potential pathogens and protect the plant by producing antibiotics, antifungal chemicals and insecticides (García-Salamanca et al., 2013). Bacillus, Pseudomonas, Enterobacter, Acinetobacter, Burkholderia and Arthrobacter are the most common microorganisms present in rhizosphere and are referred to as PGPR. Their role is to improve soil nutritional quality for better plant growth (Dennis, Miller & Hirsch, 2010; Zhang et al., 2020).
In this study, nine PGPR strains, belonging to the *Buttiauxella* and *Pseudomonas* genera, were used for the root irrigation experiment. *Pseudomonas* was the most promising group of rhizobacteria in terms of plant growth promotion, as they usually manifest a wide range of plant growth-promoting traits, such as antibiotic production, phosphate solubilization, nitrogen fixation, ACCD activity, the production of plant-beneficial compounds (plant hormones, siderophores, EPS, IAA, HCN and ammonia) and stress alleviation (*Bhattacharyya & Jha, 2012*; *Saber et al., 2015*). These PGPR of the *Pseudomonas* genus demonstrated different growth-promoting effects on blueberry cutting-seedlings. The number of branches and plant height of cutting-seedlings were significantly increased by *Pseudomonas* strains (Fig. 1). Other physiological indexes of the blueberry plants were enhanced as well (Fig. 1).

The genus *Buttiauxella*, a member of the Enterobacteriaceae family isolated from mollusks (slugs and snails), annelids (earthworms), soil and drinking water, was reported as a PGPR in 1996 (*Müller et al., 1996*). Other strains also belonging to the *Buttiauxella* genus were shown to have effects on root extension, seed germination and so on (*de Araújo et al., 2021*; *Wu et al., 2018*). Overall, physiological indexes of blueberry plants were significantly enhanced under *Buttiauxella* treatment (Fig. 1).

## Environmental element contents and microbial diversity in blueberry cutting-seedling rhizosphere soils

Nutritional elements present in plant rhizosphere soil and transformed by microorganisms are eventually utilized and absorbed during growth and development (*Liu et al., 2021*). In modern crop production systems, natural plant–microbe–soil interactions have largely been replaced with artificial fertilizer input. The consequence is that the crop varieties may have lost the ability to maintain a diverse microbiome with a decline in the sustainability of the soil system (*Pérez-Jaramillo, Mendes & Raaijmakers, 2016*). Thus, PGPR are crucial in providing the soil with nutritional elements for plant growth (*Shabaan et al., 2022*). In this study, element contents in rhizosphere soil were increased following treatments with the selected PGPR (Fig. 2). Inoculation with PGPR significantly increased OCC in rhizosphere soil compared with the control. In addition, most PGPR isolates also increased the TNC, APHC and APOC in rhizosphere soil. The PGPR had great potential and could act as a commercial biofertilizer by solubilizing minerals (*Rahimi et al., 2020*). PGPR could also improve microbial community structure for better soil quality and sustainable soil cultivation (*Bhattacharyya & Jha, 2012*). In this study, there were significant differences in the microbial diversity between rhizosphere soil samples collected from blueberry cutting-seedlings under different treatments. All PGPR isolates increased the percentage of *Occallatibacter* and *Pseudomonas* compared with the control (Fig. 3A). The percentages of *Ascitendus* and *Thysanorea* in cutting-seedlings rhizosphere soil samples inoculated with PGPR were significantly higher than those in the control soil samples (Fig. 3B). These changes in the blueberry cutting-seedlings rhizosphere soil treated with PGPR might represent a direct or indirect method of increasing crop yields and promoting plant growth (*Vejan et al., 2016*).

### Correlations between PGPR strains, rhizosphere microenvironmental factors and physiological indexes of blueberry plants

The plant rhizosphere is a complex environment that can significantly affect plant growth. As an important rhizosphere environmental factor, the rhizomicrobiome is an effective nutrition source and plays key roles in promoting plant growth. Research has demonstrated that inoculating plants with PGPR could represent an effective strategy to stimulate crop growth. The PGPR evaluated in this study could promote the growth of blueberry cutting-seedlings, increase the photosynthetic rate, and accelerate the growth of above-ground parts and roots (Fig. 1). Compared with the element contents and physiological indexes in the control and treatment groups, the dominant phyla in the soil samples had the greatest correlation with the phosphorus solubilization and auxin production of PGPR strains (Table 1, Appendix Tables 2 and 4). The altered rhizosphere microbial community structure led to changes in the soil element contents, which promoted the growth of plants (Table 2, Appendix Tables 3 and 5). The results of PCA showed that the accumulating contribution rate of these five main principal components was 86.98%. The phosphorus-solubilizing production of these strains and the rhizosphere microbial community structure were important factors representing the link between the phosphorus-solubilizing ability and auxin-production ability of PGPR strains with rhizosphere microenvironmental factors and plant growth status.

The growth-promoting bacteria themselves and the rhizosphere microbial community also significantly affect plant growth. A large range of microbial metabolites, and physical signals that trigger cell–cell communication and appropriate responses were transported between PGPR and microbial populations inside the rhizosphere soil (*Besset-Manzoni et al., 2018*). Microbial diversity is the most significant factor influencing nutrient elements in soil (*Song et al., 2021*), and soil nutrient limitations represent a major environmental condition that reduces plant growth, productivity and quality (*Gong et al., 2020*). Therefore, using PGPR with different growth-promoting effects to improve blueberry plant soil nutrients can not only promote plant growth, but also prevent the negative effects of artificial fertilizer on the soil and environment.

## CONCLUSION

In this study, blueberry plant growth was promoted by irrigating the rhizosphere with nine growth-promoting rhizobacteria strains belonging to the genera *Buttiauxella* and *Pseudomonas*. The rhizospheric microenvironment and soil nutrients demonstrated a close relationship with these strains. Generally, the rhizosphere soil microbial community structure was changed following root PGPR irrigation, where the rhizosphere soil elements that are beneficial to plant growth were increased. The results of this study are useful for the development of a rhizosphere-promoting microbial fertilizer to increase blueberry plant growth.

### Funding

This study was funded by the Natural Science Basic Research Program of Shaanxi (2021JQ-755), and 2022 College Students' Innovation and Entrepreneurship Training Program of Shaanxi University of Technology (S202210720072). The funders had no role in study design, data collection and analysis, decision to publish, or preparation of the manuscript.

### Grant Disclosures

The following grant information was disclosed by the authors:
Natural Science Basic Research Program of Shaanxi: 021JQ-755.
2022 College Students' Innovation and Entrepreneurship Training Program of Shaanxi University of Technology: S202210720072.

### Competing Interests

The authors declare that they have no competing interests.

### Author Contributions

- Mengjiao Wang conceived and designed the experiments, performed the experiments, analyzed the data, prepared figures and/or tables, authored or reviewed drafts of the article, and approved the final draft.
- Xinlong Yang performed the experiments, analyzed the data, authored or reviewed drafts of the article, and approved the final draft.

### Data Availability

All the raw measurements are available in the figures and tables and appendix.

The 16S rDNA sequence of important microbes are available at GenBank: *Pseudomonas umsongensis* (MW407040); *Pseudomonas extremorientalis* (MW407037); *Pseudomonas koreensis* (MW407038); Pseudomonas reinekei (MW407039); Pseudomonas sp. (MW407036); Buttiauxella sp. *Buttiauxella brennerae* (MW407042); Buttiauxella sp. (MW407041); *Buttiauxella gaviniae* (MW407043).

The results of high-throughput sequencing are available at Sequence Read Archive: PRJNA983920.

### Supplemental Information

Supplemental information for this article can be found online at http://dx.doi.org/10.7717/peerj.16992#supplemental-information.

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
