# Peer review of "Effects of plant growth-promoting rhizobacteria on blueberry growth and rhizosphere soil microenvironment"

_PeerJ, doi:10.7717/peerj.16992_

## Round 0.1 · original submission · Minor Revisions

The authors has to be improved the manuscript. Overall, the manuscript is good and interesting for the researchers. The reviewers has highlighted a few minor comments for the manuscript.

Reviewer 1 ·

Basic reporting

The article described the use of 9 PGPRs (plant growth promoting rhizobacteria) as a biostimulant of blueberry cuttings growth by improving rhizospheric soil nutrients and microenvironment with the modification of rhizospheric soil microbial community.
The article includes is clear and unambiguous. The introduction and background are sufficient to demonstrate how the work fits into the broader field of knowledge. The literature is relevant and is appropriately referenced (39 references).
1. Line 58 : Blueberries are a popular fruit with health benefits of the prevention of common chronic diseases (Tobar-Bolaños et al., 2021). Please add “possible health benefits instead of “health benefits”
2. Line 102 : The author must describe how the blueberry plants were prepared : “ Blueberry cutting-seedlings with five leaves and 10 cm of height were chosen for root irrigation. »
3. Line 106 : The author must indicate the optic density used for each PGPR strain : “ Every liquid beef extract peptone medium was diluted by using sterilized water to a final 50 ml volume.
4. Line 133. This sentence is not correct: “soil samples were determined using the methods can be found in Wang et al. (2021)”
The raw data shared were complete. Alla tables and figures were correct.

Experimental design

The Research questions are well defined, relevant and, meaningful. However, somes details are missing, in method's description the number of replicates is not indicated :

1. Line 96 in Plant root irrigation description, the number of repetitions was not described.
2. Line 106 “Twenty cutting seedlings were irrigated every 6d with prepared liquid beef extract peptone medium” The author hasn’t described the number of irrigations per experiment.

Validity of the findings

The results were well described, and all underlying data have been provided;
robust, statistically sound, and, controlled.
The conclusions are well stated, linked to original research question, and limited to
supporting results. However, the conclusion regarding the use of the strains must be changed:

1. Line 349 “In general, the isolated strains in our study could be used as a natural microbial fertilizer instead of traditional chemical fertilizer to promoting blueberry growth and maintain the stability of plant rhizosphere”. The author cannot do such conclusion because he has not done any comparison with chemical fertilizers

Additional comments

I recommend the author to add the strain's genius or class in the abstract and the conclusion.

·

Basic reporting

The work of Wang and Yang analyzes eight PGPR strains on the growth and some biochemical parameters of blueberry plants. In my opinion, the work is well-structured and analyzed. Although I am not an expert in editing English, it seems to me that the article lacks fluency. In addition, I have some minor suggestions:

Introduction is very short and superficial; Pease modify and write a more critical Intro. Include more recent references.
Please modify the title, try not to use ¨growth¨ word two times.
Describe in the abstract PGPR
Line 59, please re-writte the phrase ¨As most plants,¨ . I think all plants suffer from environmental effects.
L61 avoid using etc. Write the examples.
L 64, ¨microorganisms¨?
L72, cheap PGPR, why? please explain or eliminate the word.
M&M
Please include a piece of brief information about the selected strains, not only in SM.

Results
Could you do a principal component analysis in addition to the correlation analysis (table 2)? I think this figure/analysis would sum to the manuscript.

FIGURES

Figure 1, please double check the statistics, why do you have a ¨c¨ letter above ¨e¨ letters? see panel C, for example.
Please move the controls to the left, Figures 1 and 2.
Figure 5 could be improved (use biorender to draw the plant, for example). If authors want to include it, they need to give more detail from their results. Otherwise, the figure is useless and must be eliminated.

Conclusion
Please use synonyms, change ¨PGPR strains¨ from one of the two sentences.

Experimental design

The experimental design is okay, with no comments.

Validity of the findings

The results are not novel; however the work is well-done and might be of interest for the readers

Additional comments

The authors need to do some changes before its acceptance

Reviewer 3 ·

Basic reporting

The research article titledEffects of growth-promoting rhizobacteria on blueberry growth and rhizosphere soil microenvironment is a well structured paper with updated literature review and background knowledge. English language mistakes are very few. References are complete. All Figures, tables and supplementary material are well structured.

Experimental design

Methodologies used are appropriate. The PGPRs investigations on horticultural crops are rare. This research directly dealt with blueberry response to PGPRs. The results are meaningful and interested.

Validity of the findings

The results are concluded well, reliable and warrants publication.

Additional comments

I accept this article for possible publication.

---

## Round 0.2 · Minor Revisions

The manuscript is almost ready to be accepted for publication. The authors have addressed the manuscript very well and the work is suitable for other researchers who are especially working on blueberry.

The number of grammar mistakes makes the paper difficult to read. As well, many commas are placed wrongly. Thus, the manuscript should be checked by a professional service before resubmission.

Further, I recommend spelling out 'Plant growth-promoting rhizobacteria' (PGPR) in the title.

**Language Note:** The Academic Editor has identified that the English language must be improved. PeerJ can provide language editing services - please contact us at copyediting@peerj.com for pricing (be sure to provide your manuscript number and title). Alternatively, you should make your own arrangements to improve the language quality and provide details in your response letter. – PeerJ Staff

·

Basic reporting

The authors have addressed all my suggestions, it can be accepted in its current form.

Experimental design

No comment

Validity of the findings

No comment

Additional comments

No comment

---

## Round 0.3 · Minor Revisions

The manuscript is ready for publication, from a scientific viewpoint. The language needs a substantial improvement. This must be checked by a professional or by someone who is well versed in English language.

---

## Round 0.4 · accepted · Accept

Previously, the authors were asked to improve the language. They have complied and there are no apparent mistakes in language.